# High-Resolution Melting assays development for discrimination of fungal pathogens causing Grapevine Trunk Diseases

**Filipe Azevedo-Nogueira**[1,2], **Ana Gaspar**[3], **Sara Barrias**[1,2], **Bruna Costa**[1], **Beatriz Silva**[1], **Cecília Rego**[3], **Paula Martins-Lopes**[1,2]*

1 Department of Genetics and Biotechnology, DNA & RNA Sensing Lab, Universidade de Trás-os-Montes e Alto Douro, Vila Real, Portugal, 2 BioISI – Instituto de Biosistemas e Ciências Integrativas, Faculdade de Ciências, Universidade de Lisboa, Lisboa, Portugal, 3 LEAF - Linking Landscape, Environment, Agriculture and Food-Research Center, Associated Laboratory TERRA, Instituto Superior de Agronomia, Universidade de Lisboa, Lisboa, Portugal

* plopes@utad.pt

## Abstract

Grapevine Trunk Diseases are a set of fungal diseases that mainly affect wood tissues of grapevine, reducing plant fitness and yield. These diseases limit grape production, so producers need to employ several management strategies to avoid great losses. Nevertheless, due to complex etiology and irregular symptom onset that can be influenced by environmental conditions, producers find it difficult to predict outbreaks and thus to implement management practices in a timely manner. Additionally, fungal infections can remain quiescent for several years, colonizing the plant tissues without symptoms, with symptoms emergence usually occurring several years later. Therefore, the identification of infected grapevines is essential for the definition of effective management strategies. The objective of this work was to design a set of assays based on High Resolution Melting that can detect and identify the most common fungal species responsible for four Grapevine Trunk Diseases. The beta-tubulin gene (*Tub2*) was selected to design an HRM assay, considering a 130 bp amplicon. The assays distinguish ten of the most common fungal species responsible for Botryosphaeria dieback, Esca, Phomopsis dieback and Eutypa dieback, based on difference curve profiles against a known reference, *Neofusicoccum parvum*. When applied to mixtures of two fungal species, this strategy allowed both the identification and relative quantification of the pathogens. The application of such an analysis to grapevines will allow the assessment of the health status of the vines so informed management practices can be applied. This will also contribute to the reduction of the potentially pathogenic fungal load by the removal of part or the whole infected plant, ultimately reducing viticulture production costs.

**Data availability statement:** If the data are all contained within the manuscript and/or Supporting information files, enter the following: All relevant data are within the manuscript and its Supporting information files.

**Funding:** This work was funded by the Project Vine & Wine PT ref. C644866286-00000011, co-financed by Portugal Recovery and Resilience Plan (RRP) and Next Generation EU Funds. The funders had no role in study design, data collection and analysis, decision to publish, or preparation of the manuscript.

**Competing interests:** The authors have declared that no competing interests exist.

## 1. Introduction

The winemaking industry was responsible for the mobilization of over 29 billion euros in 2020, with high impact in economic development and Gross Development Product (GDP) of several countries, namely China, USA, Italy, Spain, France, Portugal, South Africa, Turkey, among others [1,2]. Wine is also widely integrated in the culture of several countries, and the worldwide grape production area is mostly dedicated to wine production [1,2]. As an economic practice with high investment from producers, several strategies are implemented to avoid and reduce quality and yield losses, as well as, to increase vineyard lifespan, which severely impacts the economic income [3,4].

*Vitis vinifera* L. has several known pathologies caused by microorganisms, being regarded as one of the most affected crops by biotic stresses [3,5]. Among the several phytopathologies that affect grapevine, Grapevine Trunk Diseases (GTDs) are amongst the most common fungal pathologies, causing significant losses, thus grape producers are highly interested in finding mitigation means [3,4]. GTDs have been detected in all grape-producing countries, with distinct incidence levels and pathogens involved, that vary accordingly to the country and the wine region [4,6–8].

So far, six diseases have been recognized as GTDs, namely Esca, Botryosphaeria Dieback, Phomopsis Dieback, Black Foot Disease, Eutypa Dieback and Petri Disease. GTDs etiology is complex with several fungal pathogens being recognized within each disease [3,4,7]. As such, Esca is caused by *Phaeomoniella (Pa.) chlamydospora, Phaeoacremonium (Pm.)* spp., and several basidiomycetous species [9–11]; Petri Disease is mainly caused by *Cadophora* spp. [9,12,13]; Botryosphaeria Dieback is caused by fungal species belonging to the Botryosphaeriaceae family [14–17]; Black Foot Disease is caused by 'Nectria'-like fungal species [6,18]; Phomopsis Dieback by *Diaporthe* spp. [19,20]; and Eutypa Dieback is caused by fungal species belonging to Diatrypaceae family [21–23].

The main classification of GTDs *in planta* is through specific symptomatology. However, performing a diagnostic using exclusively symptoms can lead to misclassification of the diseases due to several factors: (i) most of these symptoms are affected by abiotic stresses [9,24]; (ii) quiescent infections may appear irregularly in the field [25–27]; (iii) symptoms may be shared among GTDs [7,28,29]; (iv) GTD-causing species may have lower potential in infection and symptomatology [16,23,30–32]; and (v) co-occurrence of GTDs is a common event [4,6,7]. The implementation of management strategies has been a recurrent tool used to reduce the impact of GTDs, mainly based on the knowledge regarding the pathologies and their symptoms, as well as their epidemiologic status. However, this is only possible when symptoms are visible, thus it is not possible to provide an early diagnosis in plants with quiescent infections, limiting the actions for GTDs mitigation [33].

The development of molecular and chemical methods for pathogen detection in agronomical systems has been quickly arising. As of this moment, new improvements in metabolite analysis for GTDs pathogen identification in grapevine have been reported, surpassing the diagnostic capacity obtained through symptoms observation or classical fungal culture methods, *e.g.*, the detection of metabolites (*i.e.,* tyrosol and

4-hydroxybenzaldehyde) produced by *Diaporthe (Di.) eres* that induce symptoms of Phomopsis dieback [34], or the detection of Botryosphaeriaceae species infection through the analysis of metabolites (Luteoethanones A and B) with phytotoxic activity [35]. However, the production of such metabolites is highly dependent on sample and environmental factors, which can be surpassed through a genetic analysis [27,36]. Several genetic methodologies have been developed to identify infection in early stages: assays based on quantitative PCR [37,38], digital droplet PCR [38,39], or Next-Generation Sequencing [40–42]. However, most of these assays are either costly or time-consuming, as they require subsequent analyses to achieve an accurate diagnosis [22,43].

High-Resolution Melting (HRM) is a post-PCR method that provides an easier analysis by detecting variation between DNA sequences based on the melting temperature of the amplicon, as it is related to both nucleotide composition and sequence length [44,45]. HRM is also very sensitive, since the analysis is obtained by measuring the fluorescence emitted by an intercalating dye when bound to double-stranded DNA [45]. HRM is a closed-tube method with reduced handling steps, while also being highly sensitive and specific, providing a fast and cost-effective method when compared to other DNA-based technologies [46–48]. This methodology has been applied to crop protection, by identification of fungal pathogenic genera in economically valuable crops, through Internal Transcribed Spacer regions analysis [49,50]. Nevertheless, due to the recommended fragment size, this *locus* does not provide enough definition to accurately identify GTD fungal species [51]. Other *loci,* which are commonly used for phylogenetic studies, may have genomic sequences well represented in several taxa and provide enough definition for species identification [52].

The aim of this work is to develop HRM assays for the identification of several common GTD-causing pathogens based on the *Tub2* gene, providing a new, fast, and cost-effective tool for GTD detection.

## 2. Materials and methods

### 2.1. Fungal DNA extraction

Twenty-eight (28) pure fungal isolates from ten (10) GTD-causing fungal species already available and characterized (previously isolated from *Vitis vinifera*), were cultured in Potato Dextrose Agar (PDA) medium for 2–4 weeks at 25ºC in the dark (S1 File). Also, 150 mg of sawdust was obtained from three (3) grapevines, where one was naturally infected by *Diplodia (Da.) seriata,* one was naturally infected by *Botryosphaeria dothidea*, and one was naturally infected by both pathogens (confirmed by PDA platting). Total DNA isolation of the isolates and the sawdust samples was performed using the CTAB (cetyl trimethylammonium bromide) (Calbiochem, Germany) method [53], with minor modifications [49]. DNA samples were checked for concentration and quality using the NanoDrop™ 1000 Spectrophotometer (Thermo Fisher Scientific, Wilmington, DE, USA). DNA integrity was checked using electrophoresis in 1.0% agarose gels in 1 × TBE (Tris-borate-EDTA) buffer. All fungal isolate samples and sawdust samples were diluted in ultrapure water to a working concentration of 10 ng/μL. Sample mixtures of 10 ng/μL extracted DNA of *Da. seriata* (isolate 124.1) and *B. dothidea* (100.3 isolate) mixtures were made with 3:1 (75% *Da. seriata* and 25% *B. dothidea*), 1:1 (50% *Da. seriata* and 50% *B. dothidea*), and 1:3 (25% *Da. seriata* and 75% *B. dothidea*) proportions, respectively.

### 2.2. Primer design

Genomic sequences of the *Tub2* gene from the studied fungal species were obtained from NCBI database [54] (S2 File) and independently aligned in Geneious R9 (V9.1.8) software (https://www.geneious.com) to obtain consensus sequences for each fungal species. Consensus sequences gaps and nucleotide alterations that were identified in less than 95% of the aligned sequences were discarded. Consensus sequences were analyzed and conserved regions with less polymorphic events were identified for primer design. Three forward primers were designed for the amplification of a *Tub2* gene fragment in all targeted fungal species. Primer reverse used in all reactions was Bt2b [55]. All primers were synthesized by Frilabo, Lda. (Portugal) and dissolved in ultrapure water (Invitrogen, Massachusetts, USA) to 10 μM.

## 2.3. High resolution melting assays

The three newly designed forward primers were used in independent assays with universal Bt2b primer [55], making up three HRM assays based on the following primer sets: HRM2Bot-Bt2b, HRM2Damp-Bt2b and HRM2Pch-Bt2b (Table 1). Reactions were run in technical triplicates, and three fungal isolates of each species were used, except for *Da. mutila* (S1 File). Assays with DNA mixtures of *Da. seriata*/*B. dothidea* and sawdust were performed in technical triplicates, with all three primer sets. In all runs, a Negative Control (NTC) was included. As a positive control, a fungal isolate of *N. parvum* was used in all assays.

For each primer set, reactions were performed in a final volume of 25 μL containing 50 ng of genomic DNA of fungal isolates and mixture samples, 0.5 μM of each primer and 12.5 μL of MeltDoctor™ HRM Master Mix (Thermo Fisher Scientific). The initial PCR amplification included a denaturation step of 95 °C for 10 min followed by 40 cycles of 95 °C for 15 s and 60 °C for 30 s. Immediately following PCR, the HRM step was performed as follow: 95 °C for 30 s, 65 °C for 1 min rising 0.3 °C/s until 95 °C which was maintained for 15 s, where the melting curves were obtained in continuous. Fluorescence data was acquired continuously throughout the incremental melting step. All reactions were performed in a StepOne™ (48-well) Real-Time PCR System (Applied Biosystems, Thermo Fisher Scientific) and the High-Resolution Melt Software v3.0.1 (Applied Biosystems, Thermo Fisher Scientific) was used to analyze the data. Melting curves were generated after normalization and temperature shift determination. Using the software, the pre-melt and post-melt temperature regions were manually selected to determine the specific melting region for each assay, from which melting curves were generated. After removing outliers, the melting curves were distributed automatically into variants [56].

## 3. Results and discussion

Consensus sequences were created for each of the studied species by aligning several annotated *Tub2* sequences obtained from NCBI nucleotide database. This gene was targeted as it is considered the most informative for GTDs species identification [7,52]. These consensus sequences were aligned to identify regions suitable for primer design (Table 1), while the amplified fragment detained polymorphic events that allowed the differentiation among the fungal species (Fig 1).

DNA was successfully extracted from all samples (S3 File), where some presented low (under 1.8) absorbance ratios ($A_{260}$/$A_{280}$ and $A_{260}$/$A_{230}$). Samples from sawdust yielded lower DNA quantity, with lower quality, shown by the consistently lower absorbance ratios obtained.

In the first approach, HRM assays were performed to understand the best temperature range to be used in the analysis (Fig 2). To achieve good melting curve definition, in the assay with the HRM2Bot-Bt2b primer set (Fig 2A), pre-melting temperature was set to 83.2–83.7ºC and post-melting temperature to 88.8–91.5ºC. For the assay with the HRM2Pch-Bt2b primer set (Fig 2B), pre-melting temperature was set to 79.2–80.0ºC and post-melting temperature to 86.5–87.2ºC and, for the assay HRM2Damp-Bt2b primer set (Fig 2C), pre-melting temperature was set to 81.1–81.6ºC and post-melting temperature to 89.0–91.1ºC. Assay HRM2Bot-Bt2b (Fig 2A) grouped the samples in 6 variants, assay HRM2Pch-Bt2b (Fig 2B) grouped the samples into 2 variants and, as for assay HRM2Damp-Bt2b (Fig 2C) 4 variants were obtained.

We observed that more sequence dissimilarities lead to distinct melting temperatures, providing a stronger discrimination power. In a unique analysis, distinct fungal species are simultaneously analyzed, influencing the output in respect

**Table 1. Primer sets used in HRM based on the *Tub2* gene.**

| Primer set | Forward primer (5'-3') | Reverse Primer (5'-3') | Predicted amplicon size (bp) |
|---|---|---|---|
| HRM2Bot-Bt2b | GACCTCGAGCCCGGCAC | ACCCTCAGTGTAGTGACCCTTGGC | 130 |
| HRM2Pch-Bt2b | GATCTTGAGCCTGGTAC | ACCCTCAGTGTAGTGACCCTTGGC | 130 |
| HRM2Damp-Bt2b | GATCTCGAGCCCGGTAC | ACCCTCAGTGTAGTGACCCTTGGC | 130 |

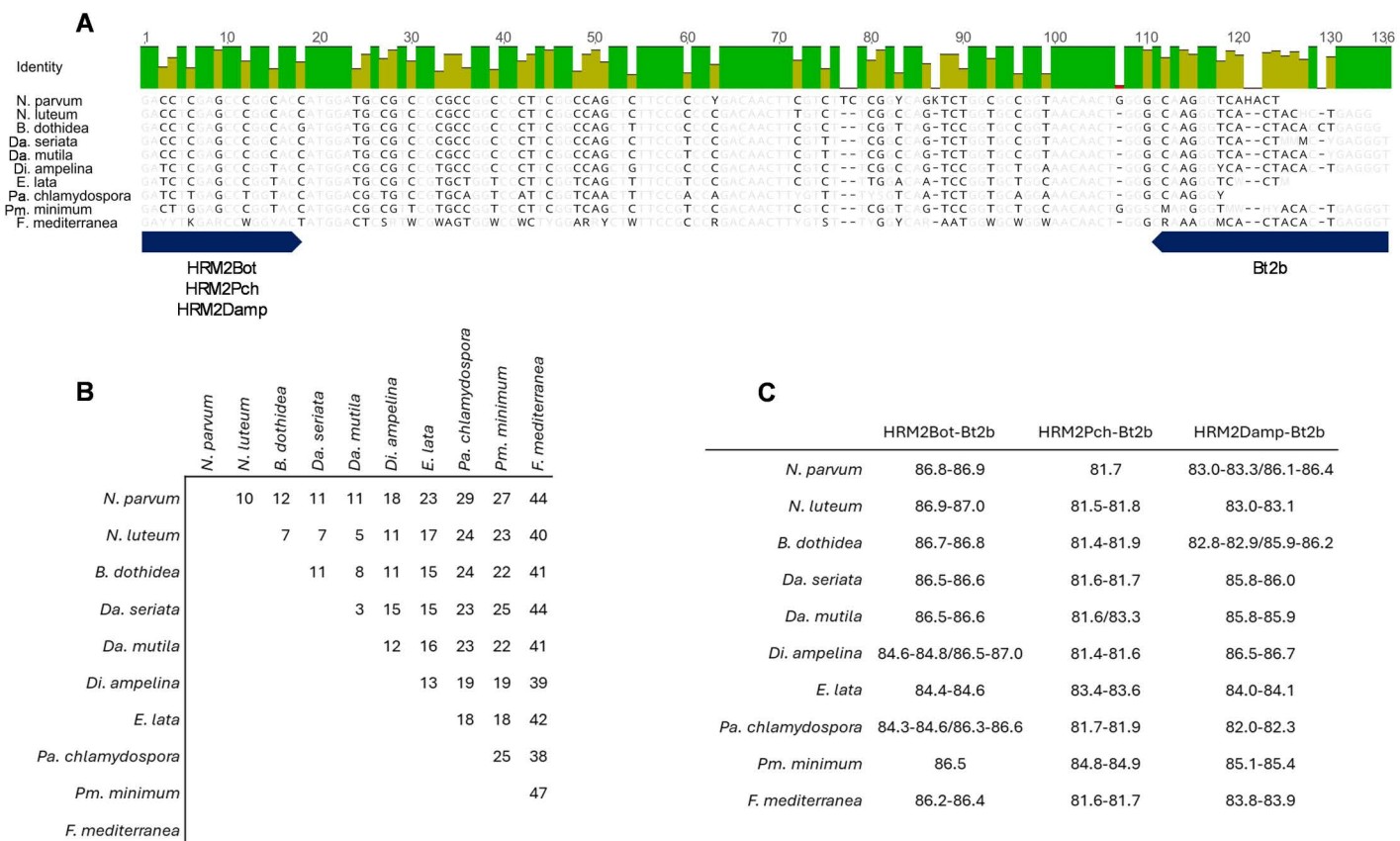

**Fig 1. Sequence and dissimilarities of the analyzed fungal species.** Alignment of the consensus sequences highlighting dissimilarities (A); matrix indicating the number of dissimilarities between the consensus sequences (B); melting temperatures of the target sequences (C).

to the variant call, leading to a less accurate identification of fungal species, namely in more closely related fungal species. Nevertheless, the annotated differences (Fig 1A) may have negligible effects on melting temperatures since some sequence alterations can camouflage others (Fig 1C). The GTDs fungal species distinction was not possible based on a unique assay, since the designed HRM assays grouped several fungal species within the same variant (Fig 2A–C). Hence, the implementation of a multi-assay analysis allowed a clear identification of the analyzed species (Fig 2D) that cause Esca, Phomopsis dieback, and Eutypa dieback. Also, one species responsible for Botryosphaeria dieback (*Neofusicoccum luteum*) presented a unique profile, being accurately identified. However, this was not observed for *Da. seriata* and *Da. mutila,* which were grouped in the same variant in all three assays. Similarly, the distinction between *N. parvum* and *B. dothidea* was not possible, since both species were grouped in the same variant in all three assays. This is an expected result, since the fungal species that are grouped together are more closely related in comparison to most of the easily distinguished fungal species, e.g., *Fomitiporia mediterranea* or *Di. ampelina* (Fig 1B), which is explained by the number of polymorphic events embedded (Fig 1) in each fungal species sequence. We observed that less related fungal species are more easily differentiated than closely related fungal species, as they have more sequence dissimilarities, which provide more discrimination power. To achieve clear identification of the more closely related fungal species, which were recurrently grouped in the same variants, we analyzed their melting curve profiles and noted that these were maintained when the assays were repeated under similar conditions, as expected. Therefore, the three assays were performed to register the melting curve profiles for each fungal species (Fig 3).

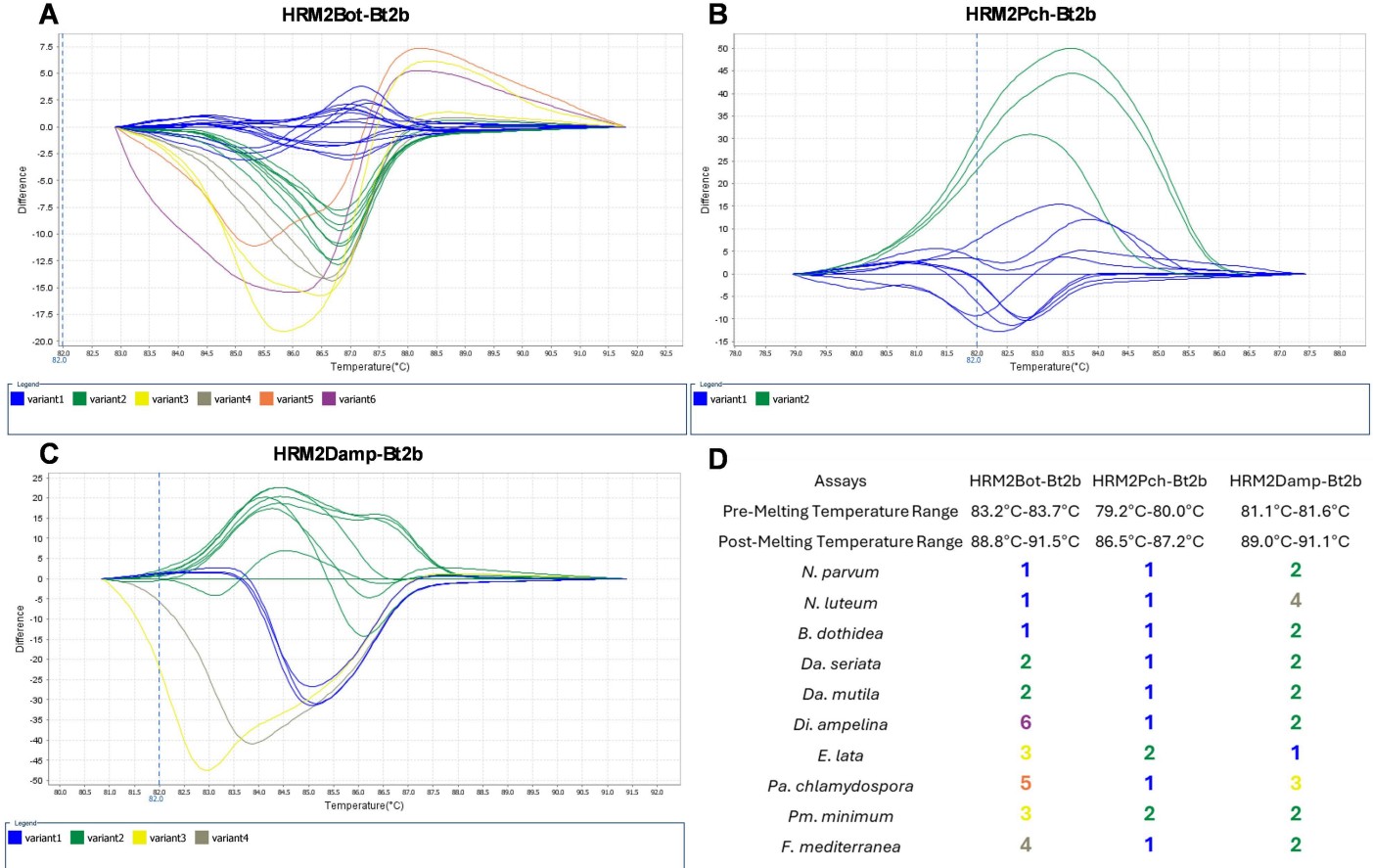

**Fig 2. HRM melting curves with three primer sets.** Different melt curves of all analyzed species with primer sets HRM2Bot-Bt2b (A) with six variants; HRM2Pch-Bt2b (B) with two variants; HRM2Damp-Bt2b (C) with four variants; and a panel with fungal species differentiation according to the variant profile (D).

For all analyzed fungal species, we observed that each species samples were grouped closely in all performed assays, due to their melting temperatures (Fig 1C) and melt curve profiles (Fig 3). However, species identification was not possible to obtain based solely on the results of one assay. As observed, melting curve profiles may be similar in one or two of the assays for the different species, *e.g., Da. mutila* and *Da. seriata* both in HRM2Bot-Bt2b and HRM2Damp-Bt2b assays. However, species classification can be achieved by applying the analysis of the three assays simultaneously, where *Da. mutila, Da. seriata*, *N. parvum,* and *N. luteum* can be successfully classified for each species, after analyzing not only the variant call, but also the melt profiles of each species.

Following, we performed the three HRM assays (Fig 4) with samples of DNA mixtures from two fungal species (*Da. seriata* and *B. dothidea*), and sawdust samples obtained from grapevine trunks naturally infected only with these pathogens. Amplification occurred in the mixture samples, however, in sawdust samples no amplification was observed (results not shown). The lack of amplification in sawdust samples can be due to the low representation of fungal DNA, since sawdust samples have a complex community (grapevine and other microbiota). Furthermore, lower absorbance ratios (S3 File) indicate a higher presence of organic contaminants, that hinder DNA amplification and subsequent analysis [57–59].

Regarding mixture samples analysis, we observed that in *Da. seriata/B. dothidea* mixture samples, the assay with the HRM2Bot-Bt2b primer set retrieved 5 variants, each representing one mixture. The assay with the HRM2Pch-Bt2b primer

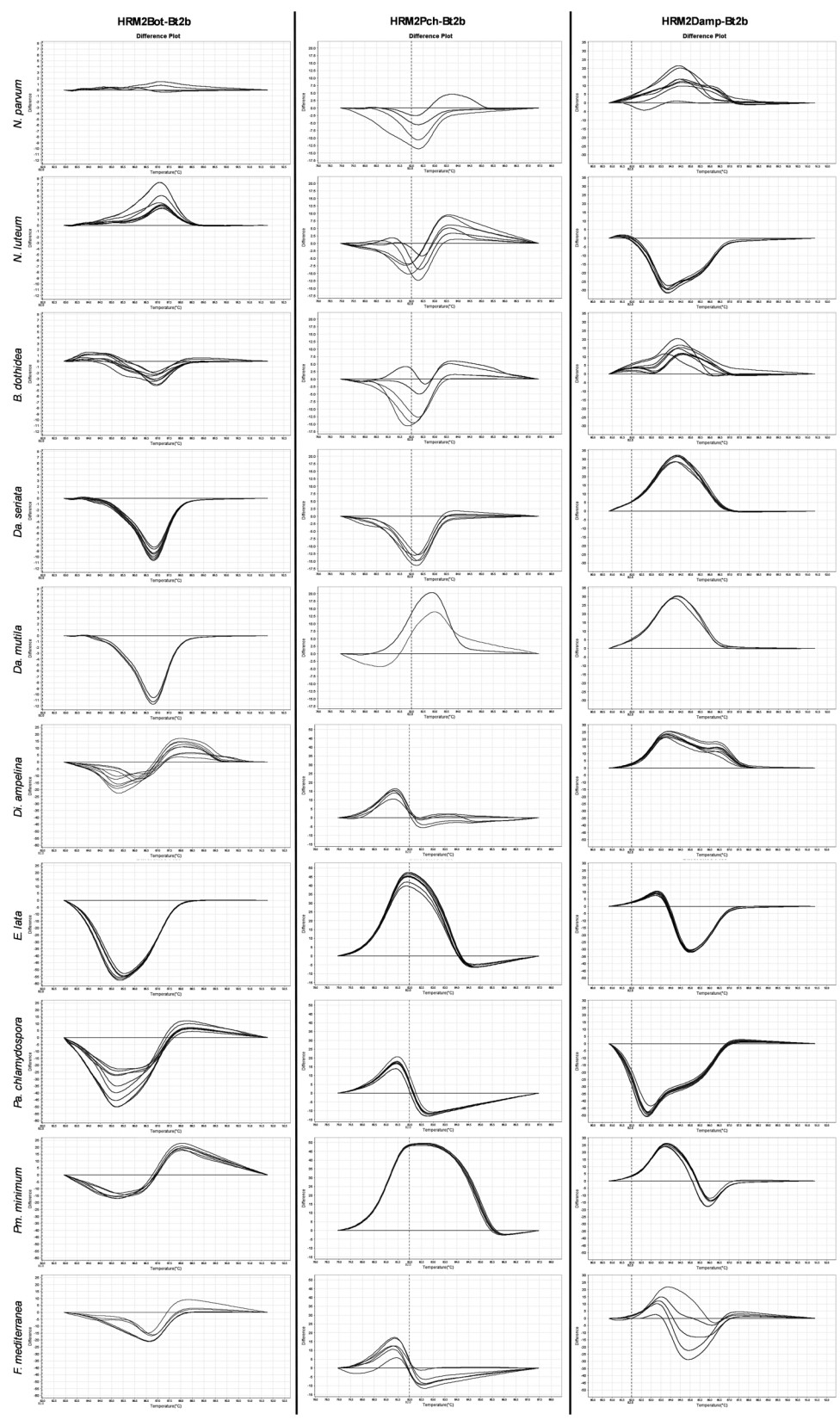

**Fig 3. In-depth analysis of the multi-assay HRM for each fungal species.** Different melting curves of the GTDs fungal species using HRM assays with primer sets HRM2Bot-Bt2b, HRM2Pch-Bt2b, and HRM2Damp-Bt2b using sample 143.1 – *N. parvum* as reference, represented by the horizontal line.

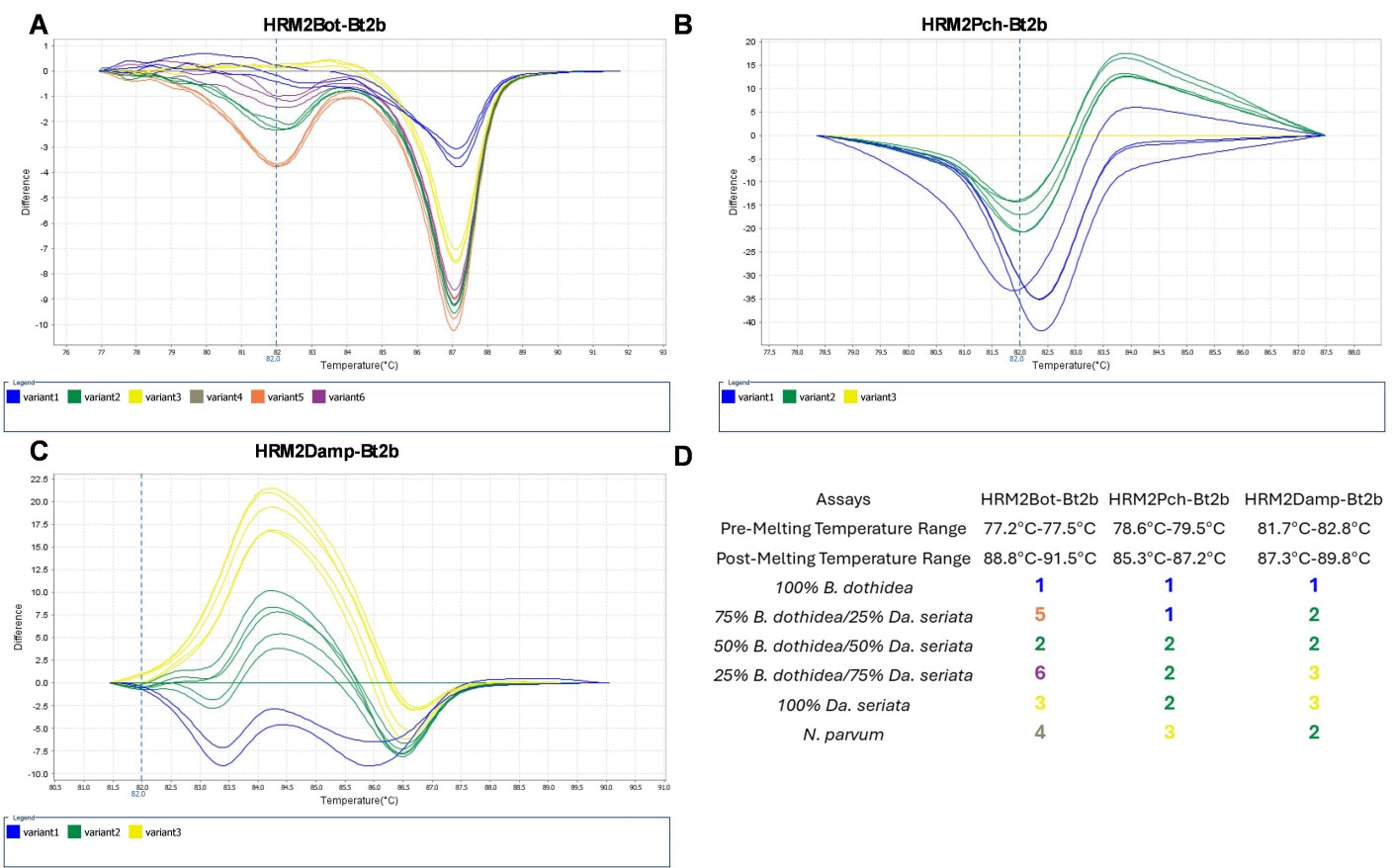

**Fig 4. Fungal species discrimination by variant clustering with the multi-assay HRM analysis in *Da. seriata/B. dothidea* mixture samples.** Difference melting curves of the *Da. seriata/B. dothidea* mixtures using HRM assays with the primer sets HRM2Bot-Bt2b assay (A) with six variants; HRM2Pch-Bt2b assay (B) with three variants; HRM2Damp-Bt2b assay (C) with three variants; and a panel with differentiation of the mixtures samples according to the variant profile (D). Isolate 143.1 (*N. parvum*) was used as reference, represented by the horizontal line.

set retrieved 2 variants, one (variant 1) englobing pure isolate sample of 100% *B. dothidea* and the mixture 25% *Da. seriata*/75% *B. dothidea*, and the other (variant 2) englobing pure isolate sample of 100% *D.seriata,* and the mixtures 75% *Da. seriata*/25% *B. dothidea* and 50% *Da. seriata*/50% *B. dothidea*. The assay with HRM2Damp-Bt2b primer set retrieved 3 variants, one (variant 1) englobing pure isolate sample of 100% *B. dothidea*, another (variant 2) englobing mixtures 75% *Da. seriata*/25% *B. dothidea* and 50% *Da. seriata*/50% *B. dothidea*, and the last (variant 3) englobing pure isolate sample of 100% *Da. seriata* and mixture 75% *Da. seriata*/25% *B. dothidea*. In all assays with the DNA mixtures, pre-melting and post-melting temperatures were adjusted to achieve the best classification of the analyzed mixtures (HRM2Bot-Bt2b primer set assay—pre-melt of 77.2–77.5ºC, post-melt of 88.8–91.5ºC; HRM2Pch-Bt2b primer set assay—pre-melt of 78.6–79.5ºC, post-melt of 85.3–87.2ºC; HRM2Damp-Bt2b primer set assay—pre-melt of 81.7–82.8ºC, post-melt of 87.3–89.8ºC).

The simultaneous analysis of mixture samples with the three assays allows for distinction of pure samples from mixture samples, allowing for greater definition when retrieving information from a mixed sample, such as complex microbial communities within a plant tissue sample. Despite being possible to identify all mixture samples with one assay (HRM2Bot-Bt2b primer set assay), the application of the other assays may allow for greater definition, since all assays are needed for species identification.

HRM is a methodology widely used to achieve a reliable result without the need to perform complex and time-consuming post-PCR procedures, such as sequencing [46,48]. Several publications have used HRM with distinct applications: medical or veterinary diagnosis [60–63]; food protection and traceability [56,64–66], and crop protection and breeding [49,67,68]. The identification of plant pathogens resourcing HRM assays has been considered useful in several cases, as for the identification of several *Phytophthora* spp. [68] and *Fusarium* spp. [69]; *Mycosphaerella graminicola*, the pathogen responsible for leaf blotch in wheat [70]; *Colletotrichum* spp., *Phytophtora* spp. and *Macrophomina phaseolina,* which affect strawberry [50]; and *Colletotrichum* spp. in olive [49]. HRM was also used to understand the endophytic fungal fraction of the wheat plant compartments (root, stem, leaf, and seeds) by targeting the Internal Transcribed Spacer (ITS) 1 and 2 regions [71] and bacterial composition of soils by targeting 16s rRNA region [72]. The use of specific primers improves this diagnostic method by simplifying results analysis and avoiding amplification of other microbes, since universal primers hybridize with several distinct fungal species, which can retrieve false positives. However, in the case of a disease with complex etiology, the use of specific primers leads to a loss of information, as several species may not amplify due to reduced primer specificity in regard to other target species, hence hampering the diagnosis [60]. The design of specific primers for GTDs may be an ineffective measure, since there are more than 150 species related to GTDs [7]. As such, the use of several primer sets/assays that amplify a small fragment in a conserved gene, commonly used for phylogenetic analysis, may be a way to unravel the presence of infection [52]. This strategy may contribute to overcoming some of the drawbacks, such as the design of several specific assays for each pathogen and avoiding loss of information by targeting the genera responsible for GTDs. Nevertheless, the ability to provide an accurate diagnosis may be hindered due to complex microbial fractions, or due to several pathogens closely related that may actuate simultaneously and in the same way, in the case of diseases with complex etiology such as observed in GTDs [4,7].

The development of three assays based on melt curve analysis by HRM allowed for the identification of several fungal species, as well as identification of two fungal species in mixture samples with distinct quantities of each isolate. This is an innovative approach for the identification of GTD-pathogens with HRM, and quantification of pathogens within mixture samples, allowing for a more integrated application of Integrated Pest Management strategies for a more sustainable viticulture. Nevertheless, some aspects must be tackled. We noted that most of the DNA samples obtained during this work presented low absorbance ratios, which are correlated with the presence of contaminants with PCR-inhibiting capacity (*e.g.,* polysaccharides and phenolic compounds) commonly present in fungal isolates [57–59]. For a reliable analysis of HRM curve profiles, it is required that PCR inhibitors must be discarded during DNA extraction, as these negatively affect fragment amplification and subsequent analysis. This is particularly relevant in GTD detection, as GTDs affect woody tissues, which is a sample matrix rich in polyphenolic compounds and polysaccharides [58]. Also, in GTDs identification, the analysis of a sampled grapevine largely colonized by an untargeted fungal species may present a drawback, since the curve profiles must be previously characterized to provide reliable identification. Furthermore, melting curve profiles can be similar between fungal species and provide a misdiagnosis, as observed regarding HRM assays using pan-specific primer pairs that targeted ITS2 region of the fungal genomes, for *Fusarium oxysporum* and *Candida lusitaniae* [63]. During this work, a set of HRM assays was developed, allowing the successful identification of ten of the most common pathogens [4,7,8]. This work sets the base for further application in vineyards management, by assessing the presence of quiescent infections and pathogen incidence, thus altering the pruning moments or chemical applications in a more efficient manner, to minimize GTDs effect on the field [4].

## 4. Conclusions

In this work, we successfully developed a set of HRM-based assays for the identification of fungal species responsible for GTDs, by amplification of a beta-tubulin gene fragment. With the proposed assays, we present a new method for GTDs detection in plant samples, providing a necessary tool for vineyards management, using a control (*N. parvum*) to normalize the curves. This work provides the basis for the development of a quick, cost-effective, and powerful tool that can help producers achieve earlier diagnosis for GTDs in field samples, thereby avoiding increased losses regarding management and mitigation strategies for these pathologies. The development of these assays, with the results integrated in a comprehensive database, provides another tool for identification and quantification of pathogen presence in grapevine wood samples.

## Supporting information

**S1 File. Grapevine Trunk Diseases fungal isolates used.**
(DOCX)

**S2 File. Accession numbers of the *Tub2* gene sequences obtained from NCBI, used for primer design.**
(DOCX)

**S3 File. DNA quantification (ng/μL), and absorbance ratios $A_{260}/A_{280}$ and $A_{260}/A_{230}$ obtained using Nanodrop™.**
(DOCX)

## Acknowledgments

The Portuguese Science and Technology Foundation (FCT), TrunkBioCode Project ref. PTDC/BAA-DIG/1079/2020 (http://doi.org/10.54499/PTDC/BAA-DIG/1079/2020), F.A.-N. fellowship (Ref. 2020.04459.BD) (https://doi.org/10.54499/2020.04459.BD) and BioISI center grants (UIDB/04046/2020 (DOI: 10.54499/UIDB/04046/2020) and UIDP/04046/2020 (DOI: 10.54499/UIDP/04046/2020), and LEAF project (reference No. UIDB/ and UIDP/04129/2020).

## Author contributions

**Conceptualization:** Cecília Rego, Paula Martins-Lopes.

**Data curation:** Filipe Azevedo-Nogueira, Sara Barrias.

**Funding acquisition:** Paula Martins-Lopes.

**Investigation:** Filipe Azevedo-Nogueira, Cecília Rego.

**Methodology:** Sara Barrias, Ana Gaspar, Bruna Costa, Beatriz Silva.

**Project administration:** Paula Martins-Lopes.

**Supervision:** Paula Martins-Lopes.

**Validation:** Paula Martins-Lopes.

**Writing – original draft:** Filipe Azevedo-Nogueira.

**Writing – review & editing:** Sara Barrias, Cecília Rego, Paula Martins-Lopes.

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
