## [Decision Letter · Decision Letter 0]

5 May 2025

Dear Dr. Martins-Lopes,

We look forward to receiving your revised manuscript.

Kind regards,

Eugenio Llorens

Academic Editor

PLOS ONE

 [This work was funded by the Portuguese Science and Technology Foundation (FCT), TrunkBioCode Project ref. PTDC/BAA-DIG/1079/2020 (http://doi.org/10.54499/PTDC/BAA-DIG/1079/2020) through the European Regional Development Fund (ERDF) through NORTE 2020 (Regional Operational Program North 2014-2020), and LISBOA 2020 (Lisboa Operational Program 2014-2020). F.A.-N. was recipient of a fellowship (Ref. 2020.04459.BD) (https://doi.org/10.54499/2020.04459.BD) from FCT (Portugal). The authors are grateful for the support of the BioISI center grants (UIDB/04046/2025, and LEAF project (reference No. UIDB /04129/2025).]. 

[This work was funded by the Portuguese Science and Technology Foundation (FCT), TrunkBioCode Project ref. PTDC/BAA-DIG/1079/2020 (http://doi.org/10.54499/PTDC/BAA-DIG/1079/2020) through the European Regional Development Fund (ERDF) through NORTE 2020 (Regional Operational Program North 2014-2020), and LISBOA 2020 (Lisboa Operational Program 2014-2020). F.A.-N. was recipient of a fellowship (Ref. 2020.04459.BD) (https://doi.org/10.54499/2020.04459.BD) from FCT (Portugal). The authors are grateful for the support of the BioISI center grants (UIDB/04046/2020 (DOI: 10.54499/UIDB/04046/2020) and UIDP/04046/2020 (DOI: 10.54499/UIDP/04046/2020), and LEAF project (reference No. UIDB/ and UIDP/04129/2020).]

 [This work was funded by the Portuguese Science and Technology Foundation (FCT), TrunkBioCode Project ref. PTDC/BAA-DIG/1079/2020 (http://doi.org/10.54499/PTDC/BAA-DIG/1079/2020) through the European Regional Development Fund (ERDF) through NORTE 2020 (Regional Operational Program North 2014-2020), and LISBOA 2020 (Lisboa Operational Program 2014-2020). F.A.-N. was recipient of a fellowship (Ref. 2020.04459.BD) (https://doi.org/10.54499/2020.04459.BD) from FCT (Portugal). The authors are grateful for the support of the BioISI center grants (UIDB/04046/2025, and LEAF project (reference No. UIDB /04129/2025).]. 

Reviewers' comments:

Reviewer's Responses to Questions

**Comments to the Author**

1. Is the manuscript technically sound, and do the data support the conclusions?

Reviewer #1: Partly

Reviewer #2: Partly

2. Has the statistical analysis been performed appropriately and rigorously?

Reviewer #1: N/A

Reviewer #2: Yes

3. Have the authors made all data underlying the findings in their manuscript fully available?

Reviewer #1: Yes

Reviewer #2: Yes

4. Is the manuscript presented in an intelligible fashion and written in standard English?

Reviewer #1: No

Reviewer #2: Yes

Reviewer #1: The manuscript " High-Resolution Melting assays development for discrimination of fungal pathogens

causing Grapevine Trunk Diseases" by Azevedo-Nogueira and co-authors presents the development of a molecular assay to discriminate between fungal species causing grapevine trunk disease (GTD). The authors demonstrate that by combining high resolution melting (HRM) profiles generated from three different primer sets targeting the beta-tubulin gene they can distinguish between a subset of fungal species. The authors finish by discussing aspects that would need to be further evaluated for the method to be applied with plant tissue. The main focus of this article is the development of a method. As currently presented the method is tested with individual fungal species used per each qPCR and HRM profiling. I think that the manuscript could be strengthened by including information of qPCR and HRM performance when using a mixed species sample and/or plant tissue. If validated with plant tissue, the tool could be applicable in the plant health and plant disease diagnostic field. I can understand the difficulty of working with trunk material and confirming infection with a single species as being a challenge for validating this assay. If that is the case alternative experiments could include spiking individual fungal DNA with plant DNA extracts and also mixing fungal cultures (and spiking with plant extracts). It remains to be determined if HRM will differentiate between the fungal species when they are present in a mixed community and with potential presence of PCR inhibitors. Also, there are sections in the manuscript that are difficult to understand. I recommend revising the manuscript text throughout, some specific examples are mentioned in the comments below (together with other comments by section of the manuscript).

Introduction

Ln 42. Change "is" to "was"

Ln 46. Change "as" to "and"

Ln 101-103. Justify the choice of the beta-tubullin as a marker

Material and Methods

Ln 107. Indicate the final number of isolates and species included in this study.

Section 2.2. Was the goal to design a universal primer that will amplify all species associated to GTDs in grape? If so, indicate this in the text. Also, how many primers were originally generated and tested (in silo and in the lab)? Or were there only three primers generated and tested (mentioned in section 2.3)? Clarify in the text.

Results

Ln 156-160. I don't think this sentence is necessary, unless this information was used to modify the qPCR and HRM assay protocols, or it influenced the HRM results. Otherwise this should be moved to the discussion.

Ln 162-169. How were the pre-melting and post-melting temperatures for the different primer tests defined?

Ln 179-197. This paragraph could use some language revision. It is difficult to read and understand. I recommend using shorter sentences overall.

Ln 192-197. This section seems more appropriate for the discussion section.

Ln 205. It is not clear to the reader what does it mean that "....the melting curve difference profiles were recurrently maintained, when assays are performed under the same conditions". I recommend rewriting this sentence to improve clarity.

Ln 215-222. This section seems more appropriate for the discussion section.

Discussion and Conclusions

Ln 227. Change "works" to "publications".

Ln 240. Remove "in the" before simultaneously.

Ln 242. Unless an in-depth analysis of results that indicate that PCR inhibitors were present in the DNA samples and that they influenced the HRM profiles is presented, I think the first two lines of this paragraph are not relevant to this work. If that information is available, however, I think it should be included and it will strengthen this manuscript.

Paragraph starting in line 242. I think before discussing the drawback of this technique, as well as future work, it is important to highlight the novel aspects of this work. For instance, the text in Ln 192-197, and lines 215-222 of the results section could be moved here. Additional tests with DNA samples of mixed cultures and DNA samples spiked with plant material could be a next step that could strengthen this work.

Figures

Figure 2A. Indicate in the figure (alignment) the regions that match the different primers.

Figure 3. Based on my understanding of the manuscript, for this figure, each individual panel represents data of three different assays. I think it could be important to use three different line colors to evaluate differences between and within assays within each figure panel. Also, in the current version of the figure, some of the lines are black, and some of the other lines are gray. There is no indication in the figure legend of what the different line color represents.

Reviewer #2: 1. Figure 3 is not clear, the letters on the chart are smeared and too small.

2. Line 93 It would be advantageous to incorporate more references into this paragraph that detail the timeline and scope of Human Resource Management (HRM) development. Additionally, it is important to summarize the findings of other researchers and illustrate how your research distinguishes itself from theirs. If your work is original and offers a unique contribution, this should be clearly articulated as well.

3. Some results are presented along with the discussion. Given that the discussion is a separate section, I recommend that these sections be reviewed.

**Do you want your identity to be public for this peer review?** For information about this choice, including consent withdrawal, please see our Privacy Policy

Reviewer #1: No

Reviewer #2: No

---

## [Author Response · Author response to Decision Letter 1]

14 Jul 2025

Reviewer 1

The manuscript " High-Resolution Melting assays development for discrimination of fungal pathogens causing Grapevine Trunk Diseases" by Azevedo-Nogueira and co-authors presents the development of a molecular assay to discriminate between fungal species causing grapevine trunk disease (GTD). The authors demonstrate that by combining high resolution melting (HRM) profiles generated from three different primer sets targeting the beta-tubulin gene they can distinguish between a subset of fungal species. The authors finish by discussing aspects that would need to be further evaluated for the method to be applied with plant tissue. The main focus of this article is the development of a method. As currently presented the method is tested with individual fungal species used per each qPCR and HRM profiling. I think that the manuscript could be strengthened by including information of qPCR and HRM performance when using a mixed species sample and/or plant tissue. If validated with plant tissue, the tool could be applicable in the plant health and plant disease diagnostic field. I can understand the difficulty of working with trunk material and confirming infection with a single species as being a challenge for validating this assay. If that is the case alternative experiments could include spiking individual fungal DNA with plant DNA extracts and also mixing fungal cultures (and spiking with plant extracts). It remains to be determined if HRM will differentiate between the fungal species when they are present in a mixed community and with potential presence of PCR inhibitors.

We performed assays with mixed fungal species samples (1) and with infected sawdust samples retrieved from infected grapevines (2). In the case of mixed species samples (1), we managed to group the fungal species according to the percentage of DNA. In the case of sawdust samples (2), we found that, with the conditions applied, it was not possible to detect the pathogens, most likely due to low fungal DNA representation within the complex sample. These results and their conclusions and limitations were added in the manuscript accordingly.

Also, there are sections in the manuscript that are difficult to understand. I recommend revising the manuscript text throughout, some specific examples are mentioned in the comments below (together with other comments by section of the manuscript).

We also altered the manuscript by fusing results and discussion sections to improve readability. This led to a Results and Discussion section, and a Conclusions section. The alterations should make the manuscript clearer.

Introduction

Ln 42. Change "is" to "was"

Ln 46. Change "as" to "and"

Altered as suggested.

Ln 101-103. Justify the choice of the beta-tubullin as a marker

A justification was added in the ll102-108.

Material and Methods

Ln 107. Indicate the final number of isolates and species included in this study.

The information was added to the manuscript.

Section 2.2. Was the goal to design a universal primer that will amplify all species associated to GTDs in grape? If so, indicate this in the text. Also, how many primers were originally generated and tested (in silo and in the lab)? Or were there only three primers generated and tested (mentioned in section 2.3)? Clarify in the text.

The goal was to design universal primers for the amplification of GTD pathogens. Since there are several fungal species associated with GTDs, design of specific primers and testing would be inviable. We designed and tested the three primers indicated in the manuscript. We indicated that in section 2.2.

Results

Ln 156-160. I don't think this sentence is necessary, unless this information was used to modify the qPCR and HRM assay protocols, or it influenced the HRM results. Otherwise this should be moved to the discussion.

We restructured the manuscript, with a merged result and discussion section. Furthermore, we have added some sawdust samples and we have commented on the presence of contaminants (as indicated by the absorbance ratios – supplementary information 3) as known to affect PCR and subsequent analysis, such as HRM, which is a possible explanation for the lack of amplification in sawdust samples.

Ln 162-169. How were the pre-melting and post-melting temperatures for the different primer tests defined?

We added this information to section 2.3.

Ln 179-197. This paragraph could use some language revision. It is difficult to read and understand. I recommend using shorter sentences overall.

We rewrote this paragraph to improve clarity, as suggested.

Ln 192-197. This section seems more appropriate for the discussion section.

We restructured the manuscript, with a merged result and discussion section. Thus, this paragraph was embedded in this section.

Ln 205. It is not clear to the reader what does it mean that "....the melting curve difference profiles were recurrently maintained, when assays are performed under the same conditions". I recommend rewriting this sentence to improve clarity.

We rewrote these sentences to improve clarity, as suggested.

Ln 215-222. This section seems more appropriate for the discussion section.

We restructured the manuscript, with the results section including discussion. Thus, this paragraph was embedded in this section.

Discussion and Conclusions

Ln 227. Change "works" to "publications".

Ln 240. Remove "in the" before simultaneously.

Altered as suggested.

Ln 242. Unless an in-depth analysis of results that indicate that PCR inhibitors were present in the DNA samples and that they influenced the HRM profiles is presented, I think the first two lines of this paragraph are not relevant to this work. If that information is available, however, I think it should be included and it will strengthen this manuscript.

We report the existence of contaminants (Supplementary Information 3) which can affect PCR and post-PCR analysis. We decided to maintain the sentences as we find this information important for the readers of the manuscript when deciding to replicate these results.

Paragraph starting in line 242. I think before discussing the drawback of this technique, as well as future work, it is important to highlight the novel aspects of this work. For instance, the text in Ln 192-197, and lines 215-222 of the results section could be moved here. Additional tests with DNA samples of mixed cultures and DNA samples spiked with plant material could be a next step that could strengthen this work

We reformulated the paragraphs and discussion to address the concern of the reviewer. Now, the discussion should be clearer with more indication of the novelty of the method.

Figures

Figure 2A. Indicate in the figure (alignment) the regions that match the different primers.

We added this information to the figure. The bioinformatic analysis indicates that all three primers bind in the same location.

Figure 3. Based on my understanding of the manuscript, for this figure, each individual panel represents data of three different assays. I think it could be important to use three different line colors to evaluate differences between and within assays within each figure panel. Also, in the current version of the figure, some of the lines are black, and some of the other lines are gray. There is no indication in the figure legend of what the different line color represents.

This figure is in a table format, in which lines represent the fungal species samples, and the columns represent the assay performed to obtain the profile curves of the fungal species. For example, the top left graph represents the profile curve of the fungal species N. parvum obtained in the assay with the primer sets HRM2Bot + Bt2b. We used only black and white in this picture to avoid confusion with other figures, since we are analyzing the curves’ profile, and not the variant call.

Furthermore, we enhance the figure to outline in black the profiles, since the gray scale was due to the identification of the sample by the program, which attributes a color for each replicate.

Reviewer 2

1. Figure 3 is not clear, the letters on the chart are smeared and too small.

The figure was enhanced to improve clarity.

2. Line 93 It would be advantageous to incorporate more references into this paragraph that detail the timeline and scope of Human Resource Management (HRM) development. Additionally, it is important to summarize the findings of other researchers and illustrate how your research distinguishes itself from theirs. If your work is original and offers a unique contribution, this should be clearly articulated as well.

We added more information to that paragraph to highlight differences between this work and other already published. To devoid of any confusion, we added information and references regarding works using High Resolution Melting (HRM), as we understand that the reviewer may have made a small mistake by indicating Human Resource Management (HRM), as the last was not in the scope of the manuscript.

3. Some results are presented along with the discussion. Given that the discussion is a separate section, I recommend that these sections be reviewed.

We also altered the manuscript by fusing results and discussion sections to improve readability. This led to a Results and Discussion section, and a Conclusions section. The alterations should make the manuscript clearer.

---

## [Decision Letter · Decision Letter 1]

12 Aug 2025

High-Resolution Melting assays development for discrimination of fungal pathogens causing Grapevine Trunk Diseases

PONE-D-25-07621R1

Dear Dr. Martins-Lopes,

We’re pleased to inform you that your manuscript has been judged scientifically suitable for publication and will be formally accepted for publication once it meets all outstanding technical requirements.

Kind regards,

Eugenio Llorens

Academic Editor

PLOS ONE

Additional Editor Comments (optional):

Reviewers' comments:

Reviewer's Responses to Questions

**Comments to the Author**

Reviewer #2: All comments have been addressed

2. Is the manuscript technically sound, and do the data support the conclusions?

Reviewer #2: Yes

3. Has the statistical analysis been performed appropriately and rigorously?

Reviewer #2: Yes

4. Have the authors made all data underlying the findings in their manuscript fully available?

Reviewer #2: Yes

5. Is the manuscript presented in an intelligible fashion and written in standard English?

Reviewer #2: Yes

Reviewer #2: This manuscript has been revised and is now ready for publication. The figure has been corrected, and the writing meets ethical standards. Both the methods and results sections are well-written, as is the discussion.

**Do you want your identity to be public for this peer review?** For information about this choice, including consent withdrawal, please see our Privacy Policy

Reviewer #2: **Yes: ** Margaretta Christita

---

## [Editor Report · Acceptance letter]

PONE-D-25-07621R1

PLOS ONE

Dear Dr. Martins-Lopes,

I'm pleased to inform you that your manuscript has been deemed suitable for publication in PLOS ONE. Congratulations! Your manuscript is now being handed over to our production team.

Kind regards,

on behalf of

Dr. Eugenio Llorens

Academic Editor

PLOS ONE